# OpenReview forum: "SEED-SET: Scalable Evolving Experimental Design for System-level Ethical Testing"
_ICLR.cc/2026/Conference — ICLR 2026 Poster_

### Official Review · Reviewer_KBvt · 2025-11-01

**Soundness:** 2
**Presentation:** 2
**Contribution:** 2
**Rating:** 4
**Confidence:** 4

**Summary:**

This paper proposes SEED-SET, a framework for ethical evaluation. The method decomposes system-level ethical assessment into a hierarchical Gaussian Process(GP) model consisting of an Objective GP $x \rightarrow y$ and a Subjective GP $y \rightarrow z$, and combines both layers within a single acquisition function that jointly accounts for mutual information and preference terms (Equation 2). The authors evaluate SEED-SET on Power Grid Resource Allocation and Fire Rescue tasks for ethical benchmarking of autonomous agents, showing that under limited sampling budgets, the method achieves an effective cost-benefit trade-off.
From a methodological standpoint, SEED-SET largely represents a structured integration of existing techniques: Variational Gaussian Processes (VGP), Bayesian Experimental Design (BED), and preference learning. Its novelty is moderate, yet introducing hierarchical modeling and a unified three-term acquisition design for ethical evaluation adds conceptual and practical value, despite relying on several handcrafted settings and strong assumptions.

**Strengths:**

1. The paper grounds subjective ethical criteria on observable outcomes through a hierarchical formulation (Objective GP $x \rightarrow y$; Subjective GP $y \rightarrow z$), rather than an end-to-end $x \rightarrow z$ mapping, which improves interpretability and sample efficiency.
2. Equation (2) formalizes the evaluation objective as a joint combination of objective information, subjective information, and a preference term, enabling an exploration-exploitation trade-off within the BED framework. The case-study results empirically support this formulation.
3. Addressing the stated challenges, "Measuring ethical behavior is difficult," "Value alignment is uncertain and dynamic" (though not fully elaborated), and "Ethical evaluation is expensive", the experiments on Power Grid and Fire Rescue demonstrate partial effectiveness.
4. Related work is comprehensive, and the overall structure is logically clear.

**Weaknesses:**

1. **Overly strong and weakly justified assumptions.** A1 fixes the policy $\pi$ and scenario space $\chi$ while assuming objective evaluations are accessible. A2 presumes users' preferences are truthful and stationary. In reality, preferences are uncertain and dynamic. These assumptions lack sufficient empirical or prior justification. Even if adopted for tractability, their rationality and limitations should be explicitly discussed, as they substantially affect the credibility of real-world ethical evaluation.
2. **Hierarchical modeling may introduce new risks.** Because subjective evaluation depends on the completeness of objective metrics, any omission or bias in these metrics may propagate to the preference layer. An end-to-end design could, in some situations, mitigate this dependency.
3. **Preference functions are manually defined.** The handcrafted weights ([0, −0.5, 1, 0] for Power Grid, [1, 0, 1] for Fire Rescue) require costly validation via TrueSkill, contradicting the stated motivation of cost efficiency and limiting scalability. The reliance on handcrafted weights weakens claims of generality.
4. **Potentially weak baselines.** It is unclear to me whether the baselines (Random, Single-GP, VS-AL-1/2) are sufficiently strong or representative of state-of-the-art Bayesian Optimization methods. More relevant or advanced baselines could strengthen the evaluation.
5. **Writing issues.**
    - Figure 1 appears early but is only explained in Section 4, with an overly brief caption.
    - The second challenge, "Value alignment is uncertain and dynamic," is not explicitly elaborated.
    - The term "both models" in the abstract is unclear until Section 4.
    - Many mathematical symbols are introduced before explanation, making it difficult for readers without a strong background to follow.

**Questions:**

1. Beyond empirical ablations, could the authors provide stronger theoretical motivation and analysis for each term in Equation (2)?
2. How generalizable is SEED-SET to other domains or multi-stakeholder settings? Can the approach maintain its performance across different ethical evaluation contexts?
3. How is the second challenge, "Value alignment is uncertain and dynamic", explicitly addressed in the proposed framework? The linkage between this challenge and the methodology remains unclear.
4. Please improve the clarity of your writing:
    - Move Figure 1 closer to Section 4 or expand its caption;
    - Define "both models" early;
    - Provide explicit explanations of symbols upon first appearance;
    - Justify Assumptions A1–A2 either empirically or via references to prior work.

**Details Of Ethics Concerns:**

The paper focuses on ethical evaluation and conducts experiments on two case studies: Power Grid Resource Allocation and Fire Rescue.
Although the authors clearly state in Section 8 that their work does not propose any new ethical norms or standards but instead aims to develop an automated framework for ethical assessment, a methodological contribution rather than a normative one, I believe that a separation between ethical evaluation tools and ethical criteria definition may carry certain risks, as the two are inherently interdependent.
While I am not an expert in ethics research and did not identify any explicit ethical issues in the paper, out of responsibility and caution, I would recommend that this paper undergo an ethics review to ensure that potential implications are properly examined.

---

> ### Author Response · Authors · 2025-11-23
> **Response to Reviewer KBvt (Part 1/3)**
>
> ## Summary
>
> Thank you for recognizing our good case study results, comprehensive literature review, and logically clear presentation!
>
> We address each of the concerns raised in this review below, and mark all changes to the revised pdf in red. Specifically we have:
> - Revised assumptions with better justification from prior works and modeling choices
> - Added BOPE baselines (four ablations) for Case-5 Power Grid case study (Appd I) to compare against our approach
> - Incorporated feedback on writing changes and mathematical symbols
> - Added additional case studies for two new problems (Optimal routing Appd E, Travel Mode Appd H), multi-stakeholder based evaluation criteria (Appd G), to show generalizability of our method.
>
>
>
> ## Details
> > **W1** Assumptions lacking sufficient empirical or prior justification.
>
> Thank you for providing feedback on the presentation of assumptions. **We have revised the all the assumptions for better readability (line 162-170), followed by justification of each based on prior literature.**
>
> We would also like to clarify that the assumption on system and policy is necessary for benchmarking, as our approach *adapts* to the system performance and generates test cases unique to each system. This is unlike static benchmarks that can be applied to any system without modification, but do not stress-test a given system sufficiently, and do not accomodate ethical criteria subjective to different users.
>
> Assumption A2 pertains to a stationary utility model for user's preference, and is a standard assumption made in preference optimization literature for tractability. **We do not assume that the preferences are deterministic, infact, our framework incorporates uncertainty from user feedback**. In practice, this allows the system to capture noisy ethical judgments while still grounding evaluation in common-sense, broadly agreed-upon ethical criteria.
>
> While assumptions A1 and A2 are made for tractability, the framework itself is flexible enough to model preference variability as the user changes.
>
> ---
> > **W2:** Hierarchical modeling may introduce new risks.
>
> We acknowledge that missing objective information can introduce testing inaccuracies. In our case studies, however, the relevant domain-specific objectives are well established in prior literature, allowing us to assume the full objective set is known a priori. This reflects real-world preference-elicitation settings, where user judgments rely on clearly defined objective metrics, and aligns with composite Bayesian Optimization formulations (e.g., Lin et al., 2022). As shown in the Power Resource study (Case-5 vs. Case-30), our hierarchical structure scales effectively to high-dimensional settings and outperforms a Single-GP end-to-end learning baseline.
>
> **From our experiments, we conclude that when the objectives are known, hierarchical learning is advantageous.** We also note that this assumption may not always hold and have updated our assumptions (Assumption A3) and limitations accordingly (Line 522-524).
>
> ---
> > **W3:** The use of TrueSkill is costly and contradicts the stated motivation of cost efficiency and limiting scalability. The reliance on handcrafted weights also weakens claims of generality.
>
> Great concern! Recall that our primary goal is to find test cases that ethically align with the user. Our results using both handcrafted weights and TrueSkill demonstrate that SEED-SET achieves better alignment than baselines.
>
> Although TrueSkill is indeed expensive to evaluate, **costly evaluation of our method does not impact scalability and cost efficiency of our method**. We simply use it to validate the alignment of our method post-training. We choose these two metrics of evaluation because in principle, it is challenging to construct unified, task-agnostic metrics to evaluate subjective preferences. Our primary contributions are in the sequential design of scenarios. Using the same argument, we claim that **using handcrafted weights also does not weaken our claims of user alignment or generality**.
>
> We also use other task specific metrics such as coverage of scenario space (Figure 6), and report other forms of visualization of preference and objective landspaces (Figure 4, Figure 16, Figure 17) to interpret the results. **We have also revised Metrics heading (Line 347-350), Section 5, to refer to additional metrics.**

---

> > ### Author Response · Authors · 2025-11-23
> > **Response to Reviewer KBvt (Part 2/3)**
> >
> > > **W4:** Potentially weak baselines.
> >
> > Thank you for your comment!
> > We have included an additional **baseline comparison based on Power Grid case study, Case-5 (Appd I) for a thorough baseline comparison with a new baseline: Bayesian Optimization with Preferennce Exploration [1])**, that deals with a specific problem setup of composite Bayesian Optimization, that can be directly compared to our method. The key difference is that BOPE decouples the training of objective and subjective models, and proposes different acquisition criteria for each. **We compare with four variants of BOPE, and show how our joint acquisition and training leads to convergence to higher preference regions, and improves on sample efficiency compared to BOPE.** We have also revised the Related Works section, Section 5.3 (Q1) to include key takeaways from the baseline comparison.
> >
> > We will also include a detailed comparison with the BOPE baselines for all our experiments in the final version.
> >
> > [1] Lin, Z. J., Astudillo, R., Frazier, P. I., & Bakshy, E. (2022). Preference Exploration for Efficient Bayesian Optimization with Multiple Outcomes. arXiv [Cs.LG]. Retrieved from http://arxiv.org/abs/2203.11382
> >
> > ---
> >
> > > **W5, Q4:** Writing concerns
> >
> > Thank you for helping us improve the clarity of our work!
> > We have carefully incorporated and addressed them as follows:
> > 1. Revised Figure 1 with a detailed caption explaining various sub-components, and included an early reference to the figure in Line 95 where we first introduce the framework.
> > 2. Revised the second challenge stated in introduction, focusing on the user-driven nature of ethical evaluation and challenges that static test cases face with the rapid evolution of autonomous systems (line 44).
> > 3. Revised 'both models' to 'learnt qualitative preferences and objectives that align with the stakeholder preferences' to introduce the high-level contribution of this work in abstract.
> > 4. Revised assumptions A1 (assumption on system) and A2 (assumptions on users), added an assumption A3 for objectives including explanation and justification for each assumption grounded in prior literature.
> > 5. Mathematical symbols: We have made revisions in Section 4.1, and Section 4.2, defining all the symbols used in the discussion.
> >
> > ---
> > > **Q1:** Beyond empirical ablations, could the authors provide stronger theoretical motivation and analysis for each term in Equation (2)?
> >
> > Thank you for this question. Below we disucss the theoretical motivation and analysis of Eq(2). **We have also included excerpts from this discussion in Section 4.2 to improve the interpretability of our key theoretical contribution with Eq (2).**
> >
> > Equation (2) reflects the inherent hierarchical structure necessary for accurately modeling ethical preferences. The subjective GP $h(y)$ can only be learned reliably if the objective GP $g(x)$ produces informative and well-resolved estimates of the ethical factors $y$. . The first term captures the expected information gain about the **objective layer**, ensuring that we reduce uncertainty in the objectives (that serve as inputs to preference modeling). The second term quantifies information gain in the **subjective layer**, directly improving our estimate of the latent utility function $h(y)$. The final term encourages sampling in regions where the current model predicts higher ethical utility, enabling the method to balance exploration with targeted exploitation.
> >
> >
> > Together, these terms arise naturally from treating ethical evaluation as a joint inference problem over both system behavior and stakeholder preferences, rather than decoupled learning stages as in traditional composite BO approaches **(see BOPE baseline discussion in Appd. I, Section 5.3 (Q1))**.
> >
> > ---
> >
> > > Q2: How generalizable is SEED-SET to other domains or multi-stakeholder settings? Can the approach maintain its performance across different ethical evaluation contexts?
> >
> > Our problem formulation is task and evaluation agnostic, and easily extends to a wide variety of evaluation settings and domains. We show this with the following new case studies in the revised version:
> > - **A multi-stakeholder evaluation case study** (Appd G) with two different evaluation criteria, showing the effect of different criteria on target objective landscape.
> > - Applying our approach to investigate alignment with real-world preference data with Travel Mode case study (Appd H)
> > - Application to **Optimal Routing case study**, where the objective is to sample routes in a city to uncover challenging scenarios for a given route planning algorithm (Appd E).

---

> > > ### Author Response · Authors · 2025-11-23
> > > **Response to Reviewer KBvt (Part 3/3)**
> > >
> > > > **Q3:** How is the second challenge, "Value alignment is uncertain and dynamic", explicitly addressed in the proposed framework?
> > >
> > > **Nonstationarity clarification:** We would like to clarify that our framework assumes stationary latent utility function $h(y)$. We understand that the second challenge 'Value alignment is uncertain and dynamic' leads to a confusion with the term dynamic, and we have revised our introduction to make our message clearer. We previously used 'dynamic' to denote that as the system capabilities evolve over time, user expectations change, and ethical alignment of a system cannot be tested adequately using static test suites and datasets without constant revisions. Our method proposes a user and system adaptive benchmarking technique, thereby automatically leading to a different dataset of scenarios for testing as the user or system changes.
> > >
> > > **Accomodating uncertainty:** Firstly, our model intrinsically allows us to incorporate uncertainty in subjective preferences using a probabilistic surrogate model to learn ethical preferences. Additionally, our subjective evaluation approach uses pairwise elicitation, which is inherently more robust to evaluation noise than absolute graded scaling, due to relative comparison, also noted in other domains [1,2]. Our preference modeling also uses a probit likelihood model $p(y_1 \succ y_2) = \Phi\bigg(\frac{h(y_1)-h(y_2)}{\sigma}\bigg)$ to capture the preference likelihood over a pair of objectives $(y_1,y_2)$ . This model also captures the uncertainty in preferences explicitly using the scaling parameter $\sigma$, where higher value of $\sigma$ denotes lesser confidence in user preferences. $\sigma$ can be adjusted to calibrate the confidence in assessment with noisy evaluations. As an example, we show LLM ablations to model, temperature and prompt for Fire Rescue case study, and observe comparable responses across all (Section 5.3 (Q5), Appd. F) This shows how our model accommodates evaluation noise using pairwise elicitation.
> > >
> > > [1] Renske Bouwer, Marije Lesterhuis, Fien De Smedt, Hilde Van Keer, and Sven De Maeyer. Comparative
> > > approaches to the assessment of writing: Reliability and validity of benchmark rating and
> > > comparative judgement. Journal of Writing Research, 15(3):498–518, 2024
> > >
> > > [2] Eva JI Hoeijmakers, Bibi Martens, Babs MF Hendriks, Casper Mihl, Razvan L Miclea, Walter H
> > > Backes, Joachim E Wildberger, Frank M Zijta, Hester A Gietema, Patricia J Nelemans, et al. How
> > > subjective ct image quality assessment becomes surprisingly reliable: pairwise comparisons instead
> > > of likert scale. European Radiology, 34(7):4494–4503, 2024

---

### Official Review · Reviewer_TT1c · 2025-11-02

**Soundness:** 3
**Presentation:** 3
**Contribution:** 2
**Rating:** 6
**Confidence:** 3

**Summary:**

This paper presents SEED SET, a method for testing ethical alignment in autonomous systems.

In this paper, an autonomous system is assumed to be a black box system that gets some input values (a scenario $x \in \mathcal X$) and outputs a set of observation values ($y\in\mathcal Y$). A stakeholder, external to the system, ranks the ethical fitness of those observations, providing a real number.

The modelling assumptions are that there exists an observation function $f_{obj}:\mathcal X \to \mathcal Y$ and an ethical evaluation function $f_{subj}:\mathcal Y \to \mathbb R$. The goal is to approximate the function $f_{subj}\circ f_{obj}:\mathcal X \to \mathbb R$, that outputs the ethical fitness of the system for each scenario.

For testing, we have different access to the observation and the ethical fitness functions. We can probe the observation function $f_{obj}$ and get noisy observation pairs $(x, y)$, where $y$ follows a Gaussian distribution centered in $ f(x)$. Given pairs of observations ($y_1, y_2$), we can probe an oracle to tell whether $f_{subj}(y_1) > f_{subj}(y_2)$ or vice versa.

To solve this problem, the paper proposes a Bayesian experimental design loop, where the objective and subjective functions are modeled as Gaussian processes, and at each time, the scenario to probe is the one that maximizes the expected gain of information.

In the experimental section, the paper reports on the implementation of this method on two case studies: a problem of resource allocation in a power grid and a problem of drone navigation in a rescue mission. As baselines, they compare with simple random allocation and more convoluted recent methods, also using Gaussian processes. They find that they are able to generate significantly more informative test scenarios than the baselines.

**Strengths:**

S1. Interesting and relevant problem for the community.

S2. Well-founded and technically sound solution.
The proposed framework builds on solid Bayesian experimental design principles and uses hierarchical variational Gaussian processes to decompose measurable system outcomes and ethical judgments. The mathematical formulation is sound, and the probabilistic modeling choices are well justified.

S3. Solid and well-executed experimental evaluation.

**Weaknesses:**

W1. The conceptual formulation of ethical alignment feels oversimplified and somewhat inconsistent with prior literature.

- The paper models ethical evaluation as a single real-valued score, whereas ethical considerations are often multi-dimensional or context-dependent, similar in structure to the scenario and observation spaces.

- In much of the ethical alignment literature, alignment is defined relationally, i.e., as the consistency between an agent's maximizing reward and an ethical ranking coming from other stakeholders.

- The use of "objective" and "subjective" terminology is confusing: it implies that ethical judgment is inherently subjective, which contradicts the assumption of a fixed function $f_{subj}:\mathcal Y \to \mathbb R$ representing a universal (a.k.a. objective) ethical ranking.

W2. The proposed solution offers limited methodological novelty relative to existing Bayesian optimization and preference-learning approaches.

**Questions:**

Q1. Instead of getting the ethical fitness of an observation as a numerical grade from the oracle stakeholder, this paper proposes a pairwise elicitation middle layer. One of the main reasons to do so is that different stakeholders may have different grading scales, so getting pairwise comparison instead of single grades is more robust. However, since you are using an LLM (the same one) for making the ethical alignment evaluation, wouldn't it be more efficient and as robust to get the grade and avoid the pairwise comparisons? Are there any other compelling reasons for this design choice?

---

> ### Author Response · Authors · 2025-11-23
> **Response to Reviewer TT1c (Part 1/2)**
>
> ## Summary
>
> Thank you for recognizing our problem is interesting, solution is technically sound, and experimental evaluation is well executed!
>
> Below, we address each of the concerns raised in the review.
> We updated the manuscript, with all changes highlighted in in red.
>
> ## Details
>
> > **Q1:** Pairwise elicitation instead of grade-based scaling?
>
> Great question!
> Grade-based scaling is known to have issues with inconsistency [1], as the expert is not aware a-priori of the complete range of observables, and the absolute confidence of assessment changes as more information is observed. Additionally, pairwise comparison also provides much higher resolution than scalar scoring, as we can compare any two pairs of objectives relative to each other, instead of categorizing them. This allows the learnt subjective model to recover a richer and more informative utility landscape.
>
> We have added LLM ablations in the revised version (Appd. F), and observe our framework is robust to LLM parameter perturbations (model, temperature, prompt), which we attribute to the relative nature of pairwise comparison. **We have also included additional references in main text (Section 5 (Q5)) that discuss likely issues with grade-based scaling in subjective evaluation.**
>
> ---
>
> > **W1.1:** Why do we model ethical evaluation as a single score?
>
> We agree that ethical considerations are inherently multi-dimensional and context-dependent. In our formulation, the latent scalar utility $z$ does not assume that ethics is fundamentally one-dimensional; rather, it represents a summary statistic of stakeholder preferences over a multi-dimensional factor space $y$. The purpose of the scalar latent utility is to provide a consistent ordinal representation of preferences that can be learned from pairwise comparisons. This approach is standard in preference learning and multi-criteria decision analysis (Lin et al 2022): heterogeneous ethical dimensions remain encoded in $y$, while the utility function $h(y)$ captures how a specific stakeholder or stakeholder group trades off among them. **Note that this also mimics realistic subjective evaluation settings, where the users are often shown certain 'metrics' (i.e, objectives) to guide their preferences, which are multi-dimensional, but the users reason over various trade-offs in their final preference (see Travel Mode case study, Appd. H).**
>
> To summarize, we approach the problem of ethical evaluation from the lens of alignment to a user-defined criteria, modeled by the utility function $h(y)$, leading to a scalar score representation.
>
> ---
>
> > **W1.2:** Relational definition of ethical alignment
>
> Our approach naturally accomodates a relational ethical alignment definition. Given a system, user(s) and ethical criteria associated to the user, our approach can be used to find scenarios that maximize ethical alignment for a given user (Power Grid case study). These scenarios can be used to measure ethical alignment. An alternate formulation is to generate scenarios that stress-test a system, and show least alignment with user's ethical criteria, which we do in the Fire Rescue case study by prioritizing scenarios with high damage values.
>
> The set of generated scenarios changes with a system, and ethical criteria, encoding the relational definition of alignment. **We also show the sensitivity of generated scenarios to user-specific ethical criteria in Appd G, in a multi-stakeholder setting, with two different criteria. (Figure 16)**
>
> ---
>
> > **W1.3:** The use of "objective" and "subjective" terminology is confusing.
>
> We use the terms objective and subjective with respect to the user(s) that define the ethical criteria. Objective refers to scenario level ethics-related measurables that can be quantified from observable features and are independent of the user. The term 'subjective' is used to refer to the user-dependent context necessary for defining the ethical alignment criteria, which makes the criteria subjective to user. **We have also made this user-specific choice of terms clearer in Introduction of the revised paper (Line 97-98).**

---

> > ### Author Response · Authors · 2025-11-23
> > **Response to Reviewer TT1c (Part 2/2)**
> >
> > > **W2:** Methodological novelty relative to literature.
> >
> > A key innovation of our method is that we **jointly update the objective and subjective models**, allowing our novel acquisition function to **simultaneously reason over both layers.**
> > In contrast, existing approaches in the BO/BED literature focusing on qualitative feedback focus on single layer pairwise elicitation settings [1] (represented by Single GP baseline in our work).
> >
> > Composite BO works [2], have investigated the composite optimization of objective and subjective models [2] but these works seperate the preference exploration (subjective modelling) and objective learning (experimentation) stages. This renders the overall framework sensitive to several design choices, leading to sub-optimal performance. The joint acquisition and learning suggested by our framework provides a natural way to perform exploration and exploitation simultaneously, without having to make these design choices.
> >
> > **We have justified this claim empirically by adding several BOPE ablations (original method proposed in [2]), and showing performance comparison with BOPE in Appd. I, Figure 18, with analysis in Section 5.3 (Q1). We have also included a conceptual analysis of Eq (2) in the main paper, which encoded our key technical contribution (Section 4.2).**  We observe that the joint acquisition and learning at each stage leads to the discovery of higher preference regions, and provides higher sample efficiency compared to baselines.
> >
> > **Therefore, to the best of our knowledge, our work is the first to fuse objective and subjective uncertainty into a joint sampling criterion for system-level ethical testing.** We have also included a discussion of these methods in Appd A.1 in the extended Literature Review.
> >
> > ---
> > [1] Chu, W., & Ghahramani, Z. (2005). Preference learning with Gaussian processes. Proceedings of the 22nd International Conference on Machine Learning, 137–144. Presented at the Bonn, Germany. doi:10.1145/1102351.1102369
> >
> > [2] Lin, Z. J., Astudillo, R., Frazier, P. I., & Bakshy, E. (2022). Preference Exploration for Efficient Bayesian Optimization with Multiple Outcomes. arXiv [Cs.LG]. Retrieved from http://arxiv.org/abs/2203.11382

---

### Official Review · Reviewer_uU6D · 2025-11-07

**Soundness:** 2
**Presentation:** 3
**Contribution:** 2
**Rating:** 2
**Confidence:** 4

**Summary:**

The paper proposes SEED-SET, scalable framework for ethical evaluation of autonomous systems.
It models evaluation as a hierarchy, an Objective GP learns the mapping from scenario x to measurable outcomes y and a Subjective GP maps outcomes y to a latent utility z that captures stakeholder preferences.

**Strengths:**

1. Joint acquisition works. Ablations show the full objective (MI + preference) beats MI-only and Pref-only, especially in higher-D Fire-Rescue.
2. Good visualizations and discussions using practical case-studies.

**Weaknesses:**

1. Author talks about the difficulty in measuring ethical behaviour, especially the subjectiveness of ethics, but then LLM is explicitly told which objective is primary and secondary in subjective evaluation. Hence the underlying preference function is already defined and can be directly expressed as a simple linear combination of the objectives. There is nothing genuinely subjective or hidden for the GP to learn.
2. Narrow ablation coverage. Lacks robustness to LLM model/prompt/temperature changes/ LLMs inherent biases. Does not compare against composite-BO/BOPE-style baselines (Lin et al. 2022).
3. No human-rater study. LLM-proxy labels and a hand crafted ħ(y) serve as ground truth, then TrueSkill uses the same LLM’s outcomes for triangulation, which is not independent.

**Questions:**

please refer to weaknesses.

---

> ### Author Response · Authors · 2025-11-23
> **Response to Reviewer uU6D (Part 1/2)**
>
> ## Summary
>
> Thank you for acknowledging our proposed joint acquisition function, good visualizations, and discussions of practical case studies!
>
> Below, we address each of the issues raised and updated our revised pdf accordingly in red.
> Notably we:
> - **Give additional ablation experiments** ablating on LLM models, prompts, and temperature in Appendix F.
> - Add additional discussion on composite-BO/BOPE literature in Section 5, Baselines, and include a BOPE baseline in Power Grid case study to compare against (Appd. I).
> - Add a case study for inferring latent objectives for air travel preference, which we validate using real world dataset (Travel Mode dataset) in Appendix H.
>
> ## Details
>
> > **W1**: LLM is explicitly told which objective is primary and secondary in subjective evaluation. Underlying preference function is already defined and can be directly expressed as a simple linear combination of the objectives.
>
> **Do we need to define a primary and secondary objective?**
> Thank you for raising an important point about the scope of subjective evaluations! We would like to clarify that our method **does not require a primary and secondary objective specification**. This was only shown in the Power Grid study as a realistic example of user-based ethical criteria. For the Fire Rescue case study, (Figure 10), the criteria is simply stated as "*high values of chemical and fire damage are preferred,"* without specifying an order of preference.
>
> **Our method supports complex ethical alignment criteria**
>
> Our framework supports a broad category of criteria going beyond those that can be expressed as a weighted sum of objectives. **To support this claim, we have added a multistakeholder case study in the Power Grid section (App. G), and a travel preference case study (App. H)**, both of which consider different complex ethical preference. The criteria in travel preference case study is not even explicitly encoded, rather, it is presented in the form of in-context learning. In the absence of a hnadcrafted preference score, we provide qualitative visualizations in Figure 16 and 17 to show that our method is able to accomodate complex evaluation criteria well.
>
> ---
>
> > **W2:** Narrow ablation coverage. Lacks robustness to LLM model/prompt/temperature changes/ LLMs inherent biases.
>
> Thank you for bringing up this concern!
> In response, **we run additional ablation experiments** validating our approach on Fire Rescue case study, and **added the new results to App. F of the revision**.
> In particular, for five random seeds, we tested our method with:
> - 5 different temperatures (Figure (13)
> - 2 additional LLM models (Figure 14), namely o3 and o3-mini in addition to the GPT 4-o from the main paper
> - 2 variations of the original prompts (Figure 15)
>
> For all ablation studies, we found no significant quantitive difference in the preference score.
> We attribute this to the pairwise style of evaluation, which mitigates uncertainity due to inconsistent evaluation, thereby leading to robustness with evaluator noise.
>
> **We have added all ablation results to App. F and discussed key results in Section 5.3, (Q5).**
>
> To ensure we respond appropriately to the question on the robustness of our method to inherent biases, could you clarify what specific types of LLM-related biases or robustness checks you had in mind?
>
> > **W2:** Does not compare against composite-BO/BOPE-style baselines (Lin et al. 2022)
>
> We thank the reviewer for pointing to composite-BO literature. We agree that our problem setup is quite similar. **Hence, we have updated our Related Works section (Section 2) with comments on BOPE.** We cannot compare against classical composite BO works (Astudillo et al 2019) since they assume access to the utility function $h(y)$, and cannot be applied directly to  black-box settings such as ours.
>
> However, BOPE (Lin et al 2022) deals with this specific consideration, and **we have included a case study for Case-5 Power Grid example, comparing four BOPE ablations to our method (Appd. I)**. The key difference is joint learning and acquisition (our work) vs separation of learning and preference exploration (BOPE), which leads to a range of design choices and sub-optimal performance (Figure 18). We analyze the effect of these design choices in Appd I and show that unlike BOPE, our method naturally accomodates exploration and exploitation at each step, leading to discovery of higher preference regions and improved sample efficiency.
>
> **We have also updated Section 5.3 (Q1) with analysis of BOPE baselines, and revised Baselines heading (Line 340, Section 5) to reflect these changes.**
>
> We will also include BOPE baselines for all experiments in the final revised version.

---

> > ### Author Response · Authors · 2025-11-23
> > **Response to Reviewer uU6D (Part 2/2)**
> >
> > > **W3**: No human-rater study. LLM-proxy labels and a hand crafted ħ(y) serve as ground truth, then TrueSkill uses the same LLM’s outcomes for triangulation, which is not independent.
> >
> > One key motivation of our work is to reduce reliance on costly domain experts by using LLMs as subjective evaluators. Running a dedicated human-rater study for every application would demand substantial expert knowledge and cost.
> >
> > Our framework is fully compatible with a human rater study whenever the required expert knowledge is available. To validate this argument, we have added an additional case study in **App. H**, **where real human data is used to measure alignment of synthetic data generated using our method with human ethical criteria (Figure 17)**.  The goal is to infer the importance of various factors considered by different people that prefer air travel, using recorded real world data. We find that the trends reported by the generated scenarios are consistent with hypothesis models proposed in the literature (Greene et al 2003), pointing to high income and less travel time as driving factors.

---

### Author Response · Authors · 2025-11-23

$$\def\a{\color{#648FFF}{\textsf{uU6D}}} \def\b{\color{#E69F00}{\textsf{TT1c}}} \def\c{\color{#DC267F}{\textsf{KBvt}}}
$$

We thank all the reviewers for their constructive feedback. We are excited that the reviewers identify the relevance and practical impact of our problem ($\a,\b$), found our approach to be technically grounded ($\b,\c$), appreciate our experimental evaluation grounded in real-world case studies (**all reviewers**), and recognize the merit of our key contribution: a hierarchical framework for system level ethical assessment (**all reviewers**). **We believe that SEED-SET makes a significant advancement in the ethical assessment of autonomous systems, by providing a grounded, quantitative approach for benchmarking that can be applied across domains, tasks, and evaluation criteria.**

We have addressed all the comments and criticisms with writing revisions in main paper, and including additional experiments and ablations. All revisions are marked in red in the revised version. We summarize the key changes below.

---
### Case studies:
We have included the following new case studies:

1. **Appendix E**: ethical assessment in Optimal routing domain to show generalizability of our approach to different domains. ($\c$)
2. **Appendix G**: multi-stakeholder based subjective evaluation for Power Grid Resource allocation, with two different evaluation criteria, showing generalizability to complex evaluation criteria and multi-stakeholder settings ($\a,\c$).
3. **Appendix H**: using real human dataset for constructing evaluation criteria, and assessment of generated scenarios against the real world dataset ($\a$).

---
### New baseline study:

**Appendix I**: A new baseline comparison for Power Grid Resource allocation case study (Case-5) that compares four ablations of a relevent baseline from BO literature (BOPE) ($\a,\c$). This baseline comparison further validates the sample efficiency and exploration capabilities of our hierarchical framework. We will include this baseline in all experiments in the final version of the paper.

---
### LLM ablation:

**Appendix F**: Added LLM parameter ablations for Fire Rescue case study (model, temperature, prompt) to evaluate robustness of our approach to evaluator noise and perturbations ($\a$).

---
### Writing changes:
We have made revisions to the following sections in main paper:
- Revised abstract with high-level motivation rather than technique specifics ($\c$)
- Revised Figure 1 and caption with more explanation. ($\c$)
- **Section 1:** added context for the use of terms 'objective', 'subjective', and revised presentation of second challenge in ethical assesment. ($\b,\c$)
- **Section 3:** explanation of assumptions with better justifications grounded in prior works. ($\c$)
- **Section 4:** revised presentation of mathematical formulation, added a conceptual interpretation for Eq (2) for better interpretability. ($\b,\c$)
- **Section 5:** Discussion of an additional baseline, added discussion on other metrics used in the paper besides Trueskill and preference score, added (Q5), (Q6) to Section 5.3 to discuss new results. ($\a,\c$)
- **Section 6:** Updated Limitations to reflect limitations on assumptions made about access to objectives. ($\c$)

We hope that these additions and clarifications fully address the reviewers’ concerns and further demonstrate the novelty, generalizability, and practical value of SEED-SET.

---

### Author Response · Authors · 2025-12-03
**Summary of Discussion**

$$\def\a{\color{#648FFF}{\textsf{uU6D}}} \def\b{\color{#E69F00}{\textsf{TT1c}}} \def\c{\color{#DC267F}{\textsf{KBvt}}}
$$

Dear AC,

Thank you for managing the review process. Below, we provide a brief summary of the reviews, the discussion, and the revisions we made. We have responded to all reviewer comments and posted a global summary of the key changes. As no follow-up comments were received, we summarize the discussion here.

Across the reviews, the reviewers highlighted the relevance and practical impact of our problem ($\a,\b$), found our approach to be technically grounded ($\b,\c$), appreciate our experimental evaluation grounded in real-world case studies (**all reviewers**), and recognize the merit of our key contribution: **a hierarchical framework for system level ethical assessment** (**all reviewers**).

**To our knowledge, SEED-SET is the first framework to offer a grounded and quantitative approach to ethical benchmarking that generalizes across domains, tasks, and evaluation criteria.**

The primary concerns from the reviewers were regarding:
1. Baseline comparison with relevent BO literature ($\a,\c$)
2. Generalizability of the our method to complex evaluation criteria and domains ($\a,\c$)
3. Lack of human rater study ($\a$)
4. Ablations on LLM parameters ($\a$)
5. Assumptions introduced in the paper, and confusion regarding terminology ($\b,\c$).

We addressed all comments, major and minor, through revisions to the main paper and by adding new experiments and ablations, with all changes marked in red in the revised version. Specifically, we have demonstrated:
* **Generalizability** by adding two new case studies (**Appendix E, Appendix G**).
* **Baseline comparison with BO literature** by comparing our method against four ablations of a composite-BO based baseline, BOPE [1] in a case study in **Appendix I.**
* **LLM ablations** to model choice, temperature, and prompt, for one of our case studies in **Appendix F**, showing robustness in the performance scores across all ablations, which we attribute to the pairwise elicitation technique used for subjective evaluation.

**Our framework generalizes to complex ethical evaluation criteria, in multi-stakeholder settings and outperforms all four ablations of the new baseline.**

* We have also added a case study in **Appendix H** that uses real human dataset. We clarify that performing human-rater study across the breadth of tasks and domains considered in the paper would demand substantial expert knowledge and cost.
* We have also revised the main paper for a more rigorous discussion on assumptions, and improving the readability of our main methodology. The key revisions are discussed below in detail.

---
### Case studies:
To address the concerns we have included the following new case studies:

1. **Appendix E**: ethical assessment in Optimal routing domain to show generalizability of our approach to different domains. ($\c$)
2. **Appendix G**: multi-stakeholder based subjective evaluation for Power Grid Resource allocation, with two different evaluation criteria, showing generalizability to complex evaluation criteria and multi-stakeholder settings ($\a,\c$).
3. **Appendix H**: using real human dataset for constructing evaluation criteria, and assessment of generated scenarios against the real world dataset ($\a$).

---
### New baseline study:

**Appendix I**: A new baseline comparison for Power Grid Resource allocation case study (Case-5) that compares four ablations of a relevent baseline from BO literature (BOPE) [1] ($\a,\c$).

---
### LLM ablation:

**Appendix F**: Added LLM parameter ablations for Fire Rescue case study (model, temperature, prompt) to evaluate robustness of our approach to evaluator noise and perturbations ($\a$).

---
### Writing changes:
We have made revisions to the following sections in main paper:
- Revised abstract with high-level motivation, Figure 1 and caption with more explanation. ($\c$)
- **Section 1:** added context for the use of terms 'objective', 'subjective', and revised presentation of second challenge in ethical assessment. ($\b,\c$)
- **Section 3:** explanation of assumptions with better justifications grounded in prior works. ($\c$)
- **Section 4:** revised presentation of mathematical formulation, added a conceptual interpretation for Eq (2) for better interpretability. ($\b,\c$)
- **Section 5:** Discussion of an additional baseline, added discussion on other metrics used in the paper besides Trueskill and preference score, added (Q5), (Q6) to Section 5.3 to discuss new results. ($\a,\c$)
- **Section 6:** Updated Limitations to reflect limitations on assumptions. ($\c$)

In light of these revisions, we believe all concerns have been satisfactorily addressed.

---
[1] Lin, Z. J., Astudillo, R., Frazier, P. I., & Bakshy, E. (2022). Preference Exploration for Efficient Bayesian Optimization with Multiple Outcomes. arXiv [Cs.LG]. Retrieved from http://arxiv.org/abs/2203.11382

---

### Meta-Review · Area_Chair_1XSo · 2026-01-06

**Summary:**

1. Limitations in the kind of ethical objectives that can be specified.
2. No experiment with human raters.
3. Ethical concerns are modelled as a single scalar score.
4. Assumptions are too strong.


Overall, I think this paper would have average score 6 (after disregarding the bad review and taking into account my projections about the scores). That's why I am recommending acceptance.

**Reviewer Concerns:**

1. Limitations in specifying ethical objectives (fully resolved by clarifying that the systems allows for expressive objectives).
2. No human raters (resolved acceptably by saying human raters are expensive / often a bottleneck).
3. Single ethics Score. (resolved partially by saying preferences often (under regularity assumptions) map to scalar scores).
4. Assumptions (resolved partially by saying they are unavoidable and standard).

**Reviewer Scores:**

1. uU6D (still 2) (this review should likely be disregarded; low-quality reviewers rarely participate or change their scores)
2. TT1c (6->8) (authors' response to the single score concern is good)
3. KBvt (still 4) (author's response about assumptions makes sense, but is unlikely to convince the reviewer).

---

### Decision · Program_Chairs · 2026-01-26

Accept (Poster)